# Vitamin D_3_, 25-Hydroxyvitamin D_3_, and 1,25-Dihydroxyvitamin D_3_ Uptake in Cultured Human Mature Adipocytes

**DOI:** 10.3390/nu17132107

**Published:** 2025-06-25

**Authors:** Nazlı Uçar, Richard. T. Pickering, Peter M. Mueller, Jude T. Deeney, María Morales Suárez-Varela, José Miguel Soriano, Michael F. Holick

**Affiliations:** 1Section of Endocrinology, Diabetes, Nutrition and Weight Management, Department of Medicine, Boston University Chobanian & Avedisian School of Medicine, Boston, MA 02118, USA; nucar@bu.edu (N.U.); jdeeney@bu.edu (J.T.D.); 2Research Group in Social and Nutritional Epidemiology, Pharmacoepidemiology and Public Health, Department of Preventive Medicine and Public Health, Food Sciences, Toxicology and Legal Medicine, School of Pharmacy and Food Sciences, Universitat de Valencia, Avenida Vicent Andres Estelles s/n, 46100 Burjassot, Valencia, Spain; maria.m.morales@uv.es; 3Section of Preventive Medicine and Epidemiology, Department of Medicine, Boston University Chobanian & Avedisian School of Medicine, Boston, MA 02118, USA; rtpicker@bu.edu; 4CARBOGEN AMCIS AG, CH-4416 Bubendorf, Switzerland; peter.mueller@carbogen-amcis.com; 5Biomedical Research Center Network on Epidemiology and Public Health (CIBERESP), Institute of Health Carlos III, Av. Monforte de Lemos, 3-5, Pabellón 11, Planta 0, 28029 Madrid, Spain; 6Observatory of Nutrition and Food Safety for Developing Countries, Food & Health Lab, Institute of Materials Science, University of Valencia, Carrer Catedrático Agustín Escardino 9, 46980 Paterna, Valencia, Spain; jose.soriano@uv.es

**Keywords:** vitamin D_3_, 25-hydroxyvitamin D_3_, 1,25-dihydroxyvitamin D_3_, adipocytes, pre-adipocytes, obesity, vitamin D metabolism, sequestration, uptake kinetics, triglycerides

## Abstract

**Background/Objectives:** Vitamin D_3_ is predominantly sequestered in adipose tissue, where it is slowly mobilized under conditions of deficiency in vivo. However, the kinetics of its uptake, release, and interaction with its major metabolites, 25(OH)D_3_ and 1,25(OH)_2_D_3_, remain poorly understood. Given the close relationship between obesity, low-grade chronic inflammation, and disrupted vitamin D metabolism, a clearer understanding of these dynamics in adipocytes is essential. Thus, we sought to characterize time-dependent uptake and metabolites in differentiated human adipocytes. **Methods:** Human pre-adipocytes were differentiated in vitro and exposed to either vitamin D_3_ and 1,25(OH)_2_D_3_ or the combination of vitamin D_3_, 25(OH)D_3_ and 1,25(OH)_2_D_3_. Intracellular concentrations were quantified through HPLC at various time points. A separate efflux experiment assessed vitamin D_3_ release under basal and isoproterenol-stimulated conditions using ^3^H-vitamin D_3_ and scintillation counting. **Results:** Vitamin D_3_ uptake showed a gradual and sustained increase over 96 h, suggesting ongoing accumulation within lipid-rich compartments. In contrast, 25(OH)D_3_ and 1,25(OH)_2_D_3_ peaked rapidly within the first hour and declined sharply. Isoproterenol stimulation significantly enhanced vitamin D_3_ release into the extracellular medium from the adipocytes, indicating increased efflux during lipolytic activation. **Conclusions:** Adipocytes selectively retain vitamin D_3_ while rapidly clearing its hydroxylated forms. These findings highlight the distinct intracellular handling of vitamin D metabolites and suggest that tailored supplementation strategies—particularly in individuals with excess adiposity—may improve bioavailability and metabolic efficacy.

## 1. Introduction

Adipose tissue is recognized as an endocrine organ performing many physiologic roles, including functions in the storage of energy reserves, regulation of appetite, control of thermogenesis, and insulin sensitivity by secreting hormones (adipokines), cytokines, and lipids [1]. Adipose tissue stores a large proportion of the body’s vitamin D (D represents D_2_ or D_3_). Also, it is an endocrine organ that can metabolize 25(OH)D_3_ and has a vitamin D receptor permitting 1,25-dihydroxyvitamin D (1,25(OH)_2_D_3_) to regulate adipocyte maturation and cytokine production [2]. It is also known that vitamin D is associated with the lipid droplet organelle inside of a mature adipocyte [1]. However, with all of this knowledge, we have little understanding of the kinetics of vitamin D_3_, 25(OH)D_3_ and 1,25(OH)_2_D_3_ uptake in adipocytes [3].

The fat-soluble vitamin D accumulates in adipose tissue [4]. Early investigations using radiolabelled vitamin D_3_ into vitamin D deficient rats revealed that a significant proportion, approximately 50%, was retained in adipose tissue as unmodified vitamin D_3_ [5]. Subsequent kinetic studies in vitamin D replete rats, following intracardiac administration of radiolabelled vitamin D_3_, showed rapid accumulation of radioactivity in the liver and serum (notably, 70% in the liver within 20 min), whereas uptake in other tissues, such as the intestinal mucosa, kidney, bone, and muscle, occurred more gradually and eventually declined [6]. In contrast, adipose tissue displayed a slow but steady accumulation of radioactivity that persisted, with no apparent clearance, and accounted for roughly 10% of the dose after one week. Notably, vitamin D metabolism was tissue specific; while polar metabolites dominated in the liver, serum, and mucosa after one week (80% of radioactivity), vitamin D_3_ itself remained predominant in kidney, muscle, and adipose tissue (70%).

Findings from human studies supported these results. Postmortem and surgical samples confirmed vitamin D_3_ accumulation in various fat depots [7]. Using a high-sensitivity high-performance liquid chromatography (HPLC) method, Lawson et al. [8] detected considerable concentrations of vitamin D_3_ (50–100 ng/g) in perirenal, pericardial, cervical, and axillary fat. Of note, perirenal fat has been identified as a depot for brown adipose tissue (BAT) in humans [9], suggesting that vitamin D_3_ also accumulates in thermogenic fat. Furthermore, Lawson’s study indicated that stored vitamin D_3_ is mobilized during periods of deficiency, with an estimated half-life of 12 days.

Beyond being sequestered in adipose tissue, vitamin D is also localized within intracellular lipid droplets of adipocytes [10]. Studies comparing vitamin D metabolism in individuals with normal weight and those with higher body fat have shown significant differences in circulation dynamics [11].

The mechanism by which vitamin D is stored and/or processed in adipose tissue remains the subject of significant interest and debate. Several hypotheses have been proposed to explain the diminished response to vitamin D supplementation. One suggests that vitamin D, being a fat-soluble vitamin, is sequestered in adipose tissue, leaving lower amounts available in circulation [12]. Another posits volumetric dilution, whereby vitamin D is distributed across a larger adiposity mass, effectively reducing its circulating concentration [13].

Little is known about how human adipocytes respond to being exposed to vitamin D_3_ and its major metabolites in a time-dependent manner. The goal of this study was to determine the kinetics for the uptake of vitamin D_3_, 25(OH)D_3_ and 1,25(OH)_2_D_3_ in cultured mature human adipocytes.

## 2. Materials and Methods

### 2.1. Cells

Frozen stocks of primary human white adipose stromal cells (ASCs) were obtained from the Boston University Adipose Biology and Nutrient Metabolism Core.

### 2.2. Differentiation of Human ASCs

Primary human ASCs (Passage 5) were seeded in 6-well plates (5000 to 15,000 cells/cm^2^) and differentiated as previously described [14]. Briefly, the preadipocytes were grown in growth medium containing alpha-MEM with 10% FBS and antibiotics until confluent. Differentiation was induced in serum-free media using an adipogenic cocktail that included insulin (Sigma-Aldrich, St. Louis, MO, USA; I0516-5 mL), dexamethasone (Sigma-Aldrich, D4902-100MG), IBMX (Sigma-Aldrich, I5879-100MG), and rosiglitazone (Sigma-Aldrich, R2408-10MG) for seven days. The medium was then changed to a lipogenic maintenance medium (NaHCO_3_, d-biotin (Sigma-Aldrich, B4639-500MG), pantothenate (Sigma-Aldrich, P5155-100G) insulin, dexamethasone), for an additional seven days.

### 2.3. Vitamin D_3_, 25(OH)D_3_, and 1,25(OH)D_3_ Incubation

Newly differentiated human adipocytes cultured in maintenance medium were treated with either 10^−6^ M vitamin D_3_, 10^−6^ M 1,25(OH)_2_D_3_, or a combination of 10^−6^ M vitamin D_3_, 10^−6^ M 25(OH)D_3_, and 10^−6^ M 1,25(OH)_2_D_3_ in the maintenance medium using three wells per condition. The combination of vitamin D_3_, 25(OH)D_3_, and 1,25(OH)_2_D_3_ 10^−6^ M was incubated for 0, 1, 2, 6, 24, and 96 h. For vitamin D_3_ or 1,25(OH)_2_D_3_ alone, the incubation times were 1 h and 96 h. After incubation, the culture media was removed, and the cells were gently washed with 1 mL of PBS. To extract intracellular contents, 1 mL of methanol was added directly to the adherent cells, followed by an additional 1 mL of methanol, resulting in a total 2 mL of per well. Extracts from each of the three wells were collected separately each time. The methanol lysate was used for high-performance liquid chromatography (HPLC) analysis and triglyceride (TG) measurement analysis.

### 2.4. Triglyceride Quantification Using Glycerol-Based Enzymatic Assay

Cellular TG content was quantified using a modified fluorometric enzymatic assay based on the detection of glycerol. Samples (250 µL from the total of 2 mL) were first dried under a stream of nitrogen gas, and the lipid residue was resuspended in Cell Lysis Buffer (CLB, Cell Signaling Technology, Danvers, MA, USA, #9803). Following sonication to ensure complete lipid solubilization, samples were centrifuged to remove debris, and the resulting supernatants were serially diluted in CLB for downstream analysis.

A glycerol-based standard curve was prepared to allow for quantification of triglyceride equivalents. A 2.5 mg/mL stock solution of glycerol was serially diluted. All standards, diluted sample replicates, and non-triglyceride reagent controls were transferred to a black 96-well plate in technical duplicates. Each well received 25 µL of triglyceride reagent (Sigma-Aldrich, T2449), and plates were sealed with adhesive film and incubated at 37 °C for 1 h, with gentle agitation every 20 min to ensure homogeneity.

A glycine-based enzyme buffer was freshly prepared by dissolving 7.88 g of glycine (Sigma-Aldrich, G7126) in 300 mL of H_2_O (Milli-Q water), adjusting the pH to 9.1 with NaOH and adding 500 µL of 2 M MgCl_2_. The final volume was brought to 500 mL with mq H_2_0, and 580 mg of ATP disodium salt (Sigma-Aldrich, A7599) was added to yield a final concentration of ~2.35 mM ATP. Ten-milliliter aliquots of this buffer were stored at −80 °C until use. For the assay, 50 µL of the enzyme buffer was added per well, followed by the addition of glycerol kinase (Sigma-Aldrich, G8121), glycerol-3-phosphate dehydrogenase (Sigma-Aldrich, G6501), and hydrazine hydrate (Sigma-Aldrich, 225819) to ensure complete conversion of glycerol to dihydroxyacetonephosphate and NADH in the presence of NAD^+^. After 20 min of incubation at room temperature, 3 µL of NAD^+^ (25 mg/mL; Sigma-Aldrich, N6522) was added to each well. Following an additional 30 min of incubation, fluorescence was measured at an excitation wavelength of 350 nm and emission at 466 nm. TG concentrations were determined based on a standard curve and expressed as triolein equivalents.

### 2.5. High-Performance Liquid Chromatography (HPLC) Procedure

The remaining 1.75 mL of methanol lysate was used for HPLC analysis. Cell-derived methanol extracts were dried under a stream of nitrogen and reconstituted in 1 mL of 5% isopropanol (IPA) in n-hexane. Samples were centrifuged to remove cellular debris, and the resulting supernatants were dried again under nitrogen. The final residues were reconstituted in 130 µL of 5% IPA/n-hexane for injection. All solvents used were of analytical HPLC grade.

HPLC was performed on an Agilent 1100 series system equipped with a variable-wavelength UV detector. Separation was achieved using a Zorbax RX-SIL normal-phase column. The mobile phase consisted of a gradient from 5% to 25% IPA in n-hexane, with the flow rate gradually increasing from 0.6 mL/min to 1.0 mL/min. Analytes were detected at 265 nm with a minimum sensitivity threshold of 12.5 ng.

### 2.6. Vitamin D_3_ Efflux Experiment

Newly differentiated human adipocytes were incubated for 48 h with 10^−6^ M vitamin D_3_ and 100,000 CPM ^3^H-vitamin D_3_. Following incubation, cells were washed and preconditioned for 2 h in DMEM without insulin or dexamethasone. Subsequently, cells were treated for 2 h under basal conditions or 10 µM isopropanol-stimulated conditions. Samples of the media were collected and analyzed through scintillation counting. Data are presented as mean ± SEM. Statistical significance was assessed through an unpaired two-tailed Student’s *t*-test (*p* < 0.05).

### 2.7. Statistical Analysis

All data are presented as mean ± standard error of the mean (SEM), unless otherwise specified. Technical replicates (*n* = 3 per condition) were used for each experimental time point. Statistical significance was determined either through one-way ANOVA with Dunnett’s post-test to compare all groups to the 1 h time point or an unpaired two-tailed Student’s t-test for comparisons between two groups. Linear regression analysis was applied to evaluate calibration curves for HPLC and fluorometric assays, with R^2^ values reported to assess fit quality. A *p*-value less than 0.05 was considered statistically significant. Statistical analyses were performed using GraphPad Prism (v9.0). In the figure, significance is indicated as follows: * *p* < 0.05, ** *p* < 0.01, *** *p* < 0.001, **** *p* < 0.0001.

## 3. Results

### 3.1. Morphological Assessment and Viability of Adipocytes Following Vitamin D_3_ Treatment

Prior to quantitative metabolite analyses, we assessed whether treatment with vitamin D_3_ or 1,25(OH)_2_D_3_ at 10^−6^ M altered adipocyte morphology or viability. Brightfield microscopy revealed no notable morphological changes in differentiated human adipocytes following treatment with either compound for up to 96 h. The cells maintained their characteristic rounded shape and lipid droplet-containing phenotype, comparable to untreated controls.

To evaluate cytotoxicity, cell death was assessed using trypan blue exclusion and LDH release assays at 1, 24, and 96 h post-treatment. Neither vitamin D_3_ nor 1,25(OH)_2_D_3_ induced a significant increase in cell death at any time point compared to vehicle controls. These findings suggest that treatment with vitamin D_3_ and 1,25(OH)_2_D_3_ at 10^−6^ M does not compromise adipocyte integrity or viability, thus validating the use of this concentration in subsequent metabolic uptake and retention analyses.

### 3.2. Quantitative Analysis of Vitamin D Metabolites Through HPLC

To quantify intracellular vitamin D_3_ and its hydroxylated metabolites, a standard calibration curve was generated using increasing concentrations of vitamin D_3_, 25(OH)D_3_, and 1,25(OH)_2_D_3_ standards. The peak areas corresponding to each injected concentration were integrated and plotted against the known quantities, as shown in Figure 1. The resulting calibration curve exhibited excellent linearity across the tested range (R^2^ > 0.99), confirming the sensitivity and reliability of the method for quantification of cellular vitamin D_3_, 25(OH)D_3_ and 1,25(OH)_2_D_3_.

### 3.3. Fluorometric Quantification of Triglycerides

To quantify intracellular TG concentrations in differentiated human adipocytes, a fluorometric assay was employed using a glycerol standard curve. The standard curve was generated using serial dilutions of glycerol expressed in triolein equivalents, ranging from 0.01 to 0.6 mg/mL. Fluorescence intensity was measured and plotted against known concentrations, resulting in a non-linear regression curve with a strong fit (R^2^ > 0.99). The standard curve allowed for accurate interpolation of TG concentrations in experimental samples based on their fluorescence output. The use of this method provided sensitive and reliable quantification of cellular triglyceride content, which was subsequently used to normalize vitamin D_3_ and vitamin D_3_ metabolite concentrations.

Figure 2 shows the time course of TG accumulation in differentiated adipocytes from 0 to 96 h. There was no significant increase over time. This validated assay provided sensitive and reproducible quantification of intracellular triglycerides, which was essential for normalizing downstream measurements of vitamin D_3_, 25(OH)D_3_ and 1,25(OH)_2_D_3_.

### 3.4. HPLC Analysis of Vitamin D_3_, 25(OH)D_3_, and 1,25(OH)_2_D_3_ in Methanol Extracts of Adipocytes

The HPLC system separated vitamin D_3_, 25(OH)D_3_, and 1,25(OH)_2_D_3_ (Figure 3A). This system was used to quantitatively determine the presence of vitamin D_3_, 25(OH)D_3_, and 1,25(OH)_2_D_3_ in the methanol cellular extracts, as demonstrated in Figure 3B,C, at 1 and 96 h.

### 3.5. Uptake of Vitamin D_3_, 25(OH)D_3_, and 1,25(OH)_2_D_3_ over Time in Cultured Human Differentiated Adipocytes

As shown in Figure 4, vitamin D_3_ concentrations increased over time, indicating continued cellular uptake and likely sequestration within lipid-rich compartments, such as lipid droplets. In contrast, 1,25(OH)_2_D_3_ reached its highest concentration 1 h after its addition and rapidly declined to undetectable levels by 96 h, suggesting rapid utilization, metabolism, or degradation. 25(OH)D_3_ concentrations also peaked at 1 h and then quickly decreased over the first 24 h, followed by a slow decline through 96 h.

These results indicate that while adipocytes gradually accumulate vitamin D_3_, the cells rapidly acquire the highest intracellular concentrations of 25(OH)D_3_ within the first hour. Both hydroxylated metabolites declined rapidly within the first 6 h, with 1,25(OH)_2_D_3_ becoming undetectable by 96 h, while 25(OH)D_3_ stabilized and gradually decreased over the remaining incubation period.

These distinct kinetics suggest that adipocytes serve not only as passive storage sites for vitamin D_3_ but also as dynamic regulators of vitamin D metabolism, reflecting the varying solubility, receptor interactions, and metabolic fates of these compounds.

To determine whether vitamin D_3_ is metabolized to either 25(OH)D_3_ or 1,25(OH)_2_D_3,_ in the differentiated human adipocytes, adipocytes were incubated with vitamin D_3_ or 1,25(OH)_2_D_3_ alone for 1 and 96 h. The results indicated that neither 25(OH)D_3_ nor 1,25(OH)_2_D_3_ were produced from vitamin D_3_ treatment, and no vitamin D esters were detected (Figure 5C). The percentage changes in the intracellular uptake of vitamin D_3_ and 1,25(OH)_2_D_3_ at 1 h and 96 h were similar to what was observed when all three vitamin D species were incubated together. Vitamin D_3_ showed a 91.8% increase in accumulation at 96 h compared to a 71.0% decrease in 1,25(OH)_2_D_3_ concentrations.

### 3.6. Vitamin D_3_ Release

Isoproterenol stimulation significantly increased the release of ^3^H-vitamin D_3_ from adipocytes that were treated with isoproterenol compared to the basal condition, indicating that lipolysis increases the release of vitamin D_3_ from adipocytes (Figure 6).

## 4. Discussion

Our findings indicate that differentiated human adipocytes exhibit distinct intracellular handling of vitamin D_3_ and two of its hydroxylated metabolites, which may contribute to—but does not fully explain—variations in systemic vitamin D bioavailability. Further mechanistic studies, including gene expression and in vivo validation, are required to clarify these relationships.

The gradual accumulation and prolonged retention of vitamin D_3_ within adipocytes suggest that it is stored within lipid-rich compartments, likely lipid droplets, potentially limiting its availability for systemic metabolism and use. In contrast, 25(OH)D_3_ and 1,25(OH)_2_D_3_ showed rapid uptake and clearance, indicating that they are not stored but rather utilized in adipocytes. This selective accumulation and retention of vitamin D_3_ supports the hypothesis that adipose tissue acts as a long-term storage site for vitamin D_3_ and not for 25(OH)D_3_ or 1,25(OH)_2_D_3_. Furthermore, we observed that treatment of the adipocytes with isoproterenol induces lipolysis, increasing the percentage of ^3^H vitamin D_3_ in the media when compared to the adipocytes treated with control media.

This selective retention aligns with previous studies suggesting that the capacity of adipocytes to sequester vitamin D_3_ leads to lower circulating vitamin D concentrations, impacting overall vitamin D status in obese patients [15,16]. Our recent study [17] supports that vitamin D_3_ is sequestered in adipose tissue, likely within lipid droplets, reducing its availability for metabolism and utilization.

It is important to highlight that vitamin D_3_ and its metabolites play essential roles in regulating calcium ion concentrations, which are critical for bone mineralization, neuromuscular function, and cellular signaling [4,11]. By promoting intestinal calcium absorption and renal calcium reabsorption, 1,25(OH)_2_D_3_ maintains serum calcium within a narrow physiological range. The sequestration of vitamin D_3_ in adipocytes observed in this study may reduce its bioavailability, potentially impairing these calcium-dependent processes. We do not fully understand why vitamin D_3_ is sequestered in adipocytes. However, it has been speculated that storage of precious vitamin D_3_ during the summer was critically important for the evolution and survival of humans as they left their ancestral home in equatorial Africa. Exposure of hunter–gatherers to sunlight during the summer provided them with their vitamin D requirement. The unutilized fat-soluble vitamin was deposited in adipose tissue with the intent of it being released in the winter in far northern and southern latitudes when no vitamin D could be produced. Due to lack of an energy source of food during the winter, the adipocyte’s lipid droplets provided an excellent source of energy while at the same time releasing vitamin D_3_ to prevent rickets in children and metabolic bone disease in adults.

These dynamics underscore the importance of adipocyte handling of vitamin D_3_ in predicting physiological outcomes and may inform future strategies for optimizing vitamin D supplementation and therapeutic use.

In addition to their role as potential reservoirs for vitamin D_3_, adipocytes express the vitamin D receptor (VDR) and possess the enzymatic machinery required for the local metabolism of vitamin D metabolites [2]. Notably, 1,25(OH)_2_D_3_ binds to VDR, inducing the expression of CYP24A1, the enzyme responsible for 24-hydroxylation, which converts both 25(OH)D_3_ and 1,25(OH)_2_D_3_ into inactive, water-soluble metabolites destined for excretion [18]. This rapid induction of CYP24A1 likely explains the sharp decline observed for 1,25(OH)_2_D_3_ and the slower decline in 25(OH)D_3_ in our study.

These patterns may be further influenced by inflammatory mediators present in obese adipose tissue potentially exacerbating vitamin D sequestration. Chronic low-grade inflammation associated with obesity alters adipocyte function and may impair vitamin D mobilization from lipid stores. The interplay between adipocyte inflammation and vitamin D metabolism may contribute to the reduced circulating 25(OH)D concentration observed in obesity, highlighting the importance of considering both metabolic and inflammatory pathways in understanding vitamin D deficiency in this population.

### 4.1. Future Directions

Further investigation is warranted to directly assess the expression of vitamin-D-related genes in adipocytes. Specifically, quantitative PCR or immunofluorescence studies assessing VDR and CYP24A1 expression, with and without exposure to vitamin D_3_, 25(OH)D_3_, or 1,25(OH)_2_D_3_, could clarify the regulatory mechanisms underlying local vitamin D metabolism. In addition, examining the uptake and retention of vitamin D metabolites in other relevant cell types, such as bone-marrow-derived mesenchymal stem cells (BMMSCs), would provide insight into tissue-specific responses and functional implications, including the potential for vitamin D to influence osteogenesis via calcium handling.

### 4.2. Strengths and Limitations

A strength of this study is the use of direct quantification via HPLC to track intracellular concentrations of vitamin D_3_, 25(OH)D_3_, and 1,25(OH)_2_D_3_ over time in human adipocytes. However, a limitation is that some experiments were conducted using differentiation batches with variable efficiency, which may have affected lipid content and vitamin D handling. Future studies should explore intracellular localization of vitamin D metabolites and the specific transporters or binding proteins involved.

## 5. Conclusions

This study provides valuable insights into the intracellular dynamics of vitamin D_3_, 25(OH)D_3_, and 1,25(OH)_2_D_3_ in human adipocytes. The findings highlight the critical role of adipocytes in accumulating vitamin D_3_, which helps explain why obese patients require 2–3 times more vitamin D to maintain the same vitamin D status as a normal weight adult. Our data suggest that under the tested in vitro conditions, human adipocytes may not retain substantial amounts of 25(OH)D_3_ or 1,25(OH)_2_D_3_, possibly due to rapid metabolism or utilization. However, further experiments examining gene expression (e.g., VDR, CYP24A1) and longer-term dynamics are necessary to confirm this interpretation. These observations underscore the need for a better understanding of how adipose tissue influences vitamin D status through its sequestration and the release of vitamin D.

## Figures and Tables

**Figure 1 nutrients-17-02107-f001:**
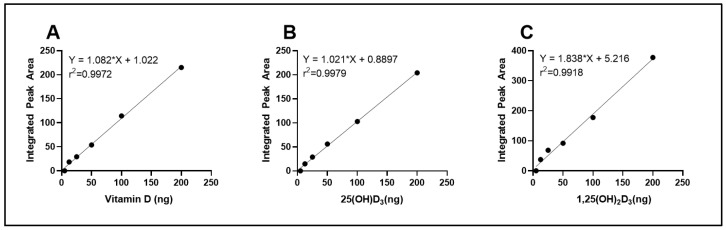
Calibration curves for quantification of vitamin D metabolites through HPLC. Standard curves were generated for (**A**) vitamin D_3_, (**B**) 25-hydroxyvitamin D_3_ [25(OH)D_3_], and (**C**) 1,25-dihydroxyvitamin D_3_ [1,25(OH)_2_D_3_] using increasing concentrations of analytical standards. The integrated peak area for each standard concentration was plotted, and linear regression was performed (R^2^ > 0.99). Copyright Michael F. Holick 2025.

**Figure 2 nutrients-17-02107-f002:**
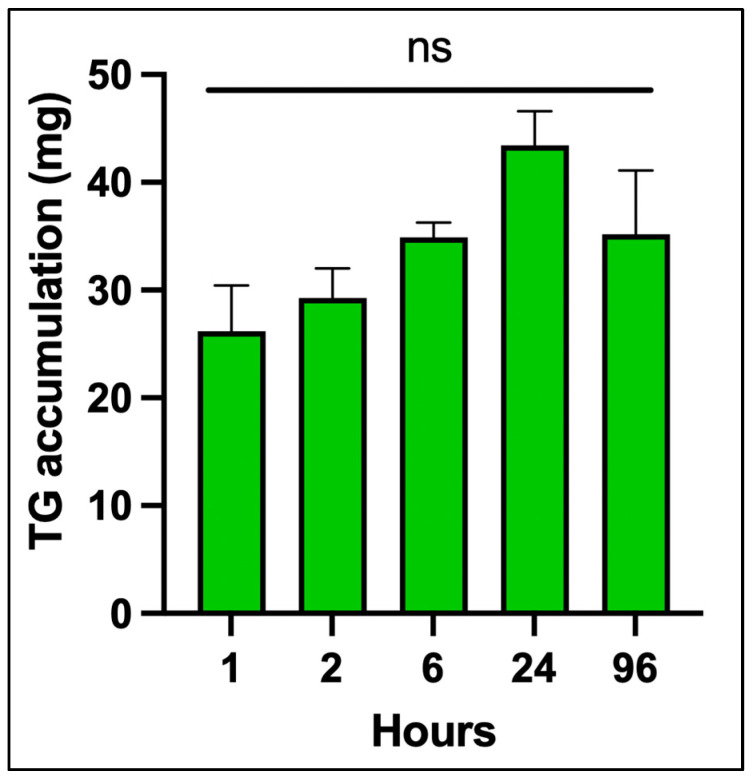
Time course of intracellular triglyceride (TG) accumulation in differentiated human adipocytes. Triglyceride content (mg TG per 2 mL) was quantified using a fluorometric assay calibrated against a glycerol-based standard curve (expressed in triolein equivalents). Differentiated adipocytes were collected at specified time points (0, 1, 2, 6, 24, and 96 h), and TG levels were interpolated based on fluorescence intensity. Values represent mean ± SD of technical replicates. Copyright Michael F. Holick 2025.

**Figure 3 nutrients-17-02107-f003:**
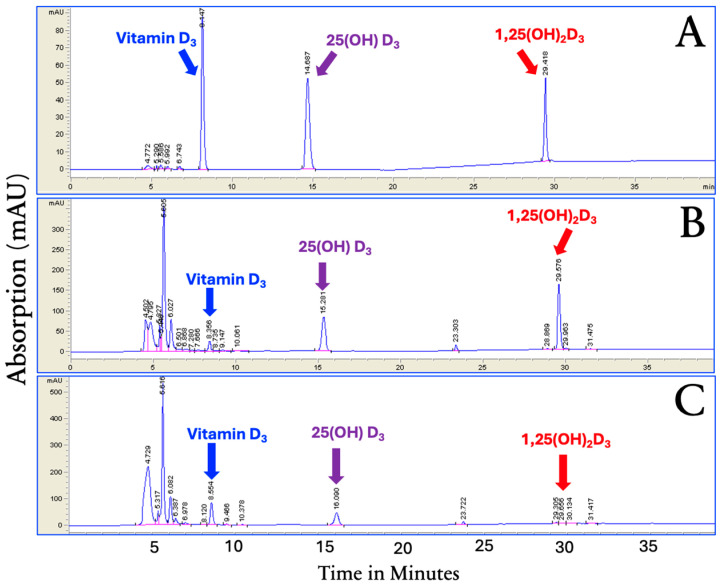
HPLC chromatograms of standards (**A**), 25(OH)D_3_, and 1,25(OH)_2_D_3_ for 1 h (**B**) and methanol extracts of human adipocytes incubated with vitamin D_3_, 25(OH)D_3_, and 1,25(OH)_2_D_3_ for 96 h (**C**). Retention times of vitamin D_3_, 25(OH)D_3_, and 1,25(OH)_2_D_3_ were consistent with standard calibration. Copyright Michael F. Holick 2025.

**Figure 4 nutrients-17-02107-f004:**
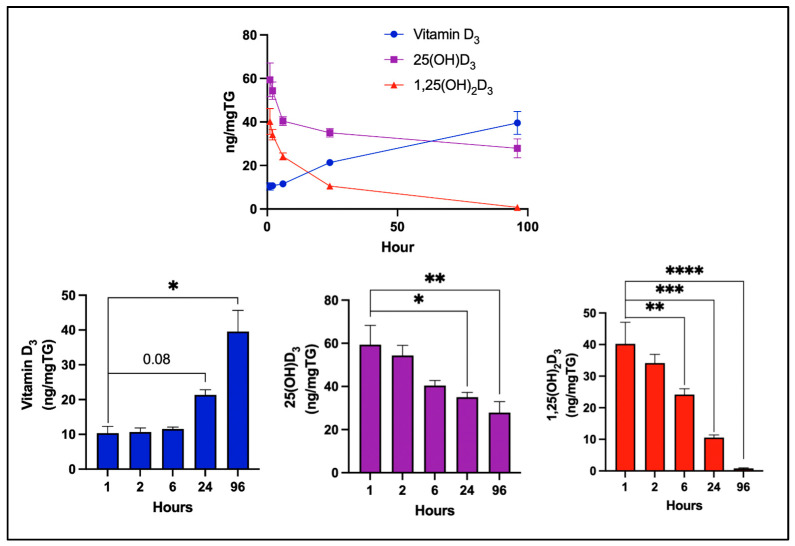
Time-dependent intracellular accumulation and decline of vitamin D_3_ and its metabolites in newly differentiated human adipocytes. Intracellular concentrations of vitamin D_3_, 25(OH)D_3_, and 1,25(OH)_2_D_3_ were quantified through HPLC over a 96 h period and normalized to total triglyceride content. The data represent the mean ± standard deviation (SD) of three biological replicates. Significance is indicated as follows: * *p* < 0.05, ** *p* < 0.01, *** *p* < 0.001, **** *p* < 0.0001. Copyright Michael F. Holick 2025.

**Figure 5 nutrients-17-02107-f005:**
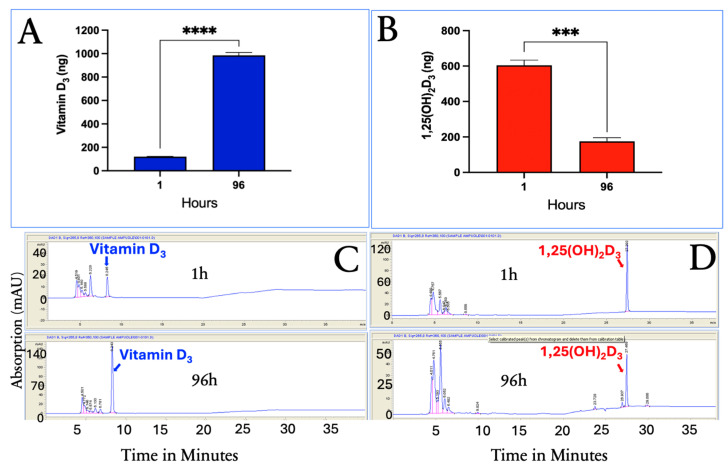
Time-dependent uptake of vitamin D_3_ and 1,25(OH)_2_D_3_ in human adipocytes. (**A**) Quantification of intracellular vitamin D_3_ levels after 1 and 96 h of incubation shows a time-dependent increase in uptake. (**B**) In contrast, 1,25(OH)_2_D_3_ levels significantly decreased over the same period. (**C**,**D**) Representative HPLC chromatograms confirming intracellular detection of vitamin D_3_ (**C**) and 1,25(OH)_2_D_3_ (**D**) at both time points. No additional vitamin D metabolites were detected. Data are presented as mean ± SEM (*n* = 3). Significance is indicated as follows: *** *p* < 0.001, **** *p* < 0.0001. Copyright Holick 2025.

**Figure 6 nutrients-17-02107-f006:**
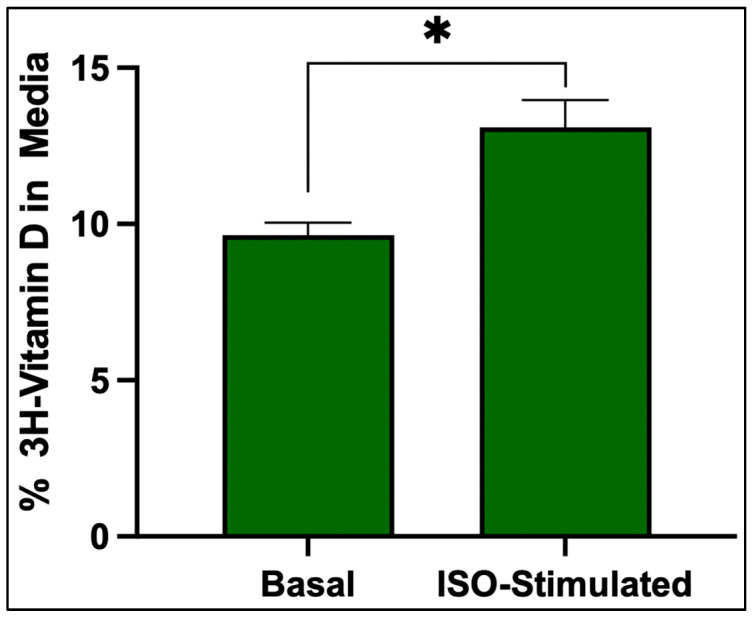
Comparison of the percent of 3H- vitamin D_3_ in the media of adipocytes that were treated with either isoproterenol dissolved in media or media control. Data are presented as mean CPM ± SEM (*n* = 3); statistical analysis was performed using an unpaired two-tailed *t*-test (*p* = 0.04). Significance is indicated as follows: * *p* < 0.05. Copyright Michael F. Holick 2025.

## Data Availability

The original contributions presented in this study are included in the article, further inquiries can be directed to the corresponding author.

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
