# Peer review of "Vitamin D3, 25-Hydroxyvitamin D3, and 1,25-Dihydroxyvitamin D3 Uptake in Cultured Human Mature Adipocytes"

_nutrients, 2025, doi:10.3390/nu17132107_

Round 1
Reviewer 1 Report
Comments and Suggestions for Authors
This study investigated Vitamin D3 and its metabolites and functional form 25(OH)D3 and 1,25(OH)2D3 uptake by human in vitro differentiated adipocytes and found a gradual and sustained increase of Vitamin D3 over 96 hours, indicating accumulation within lipid-rich compartments. In contrast, 25(OH)D₃ and 1,25(OH)₂D₃ peaked rapidly within the first hour and declined sharply. Overall the manuscript addressed an important question of adipose cells in regulation of Vitamin D and its metabolites and try to explain why obesity people need 2-3 times of Vitamin D. The manuscript is well written. I have the following comments.
- Abstract: Authors said “Isoproterenol stimulation significantly enhanced intracellular retention of vitamin D₃, indicating reduced efflux during lipolytic activation”. But the results said Isoptoterenol stimulation increased secretion (efflux) of media Vitamin D3 due to lipolysis. This is contradict. Please check.
- Statistical analysis should be a standalone section.
- Figure 6, Can authors put statistical significance on the bar, so that it is more straightforward.
- It will be interesting to test accumulation in BMMSCs in vitro uptake of Vitamin and its metabolites as these vitamin Ds will promote osteogenesis by increase Absorbance of calcium.
- Authors discussed the limitations of the study. Authors stated the rapid uptake and clearance of 25(OH)D3 and 1,25(OH)2D3 was because adipocytes have receptors and stimulate some metabolism in the adipocytes. Can author perform some Q-PCR analysis to test if the Adipocytes have Vitamin D receptor and if with and without Vitamin D3, 25(OH)D3 and 1,25(OH)2D3 can change VDR and if intracellular enzyme (CYP24A1) was actually activated during this process using PCR or IF. Traditionally concept was 1,25(OH)2 D3 work in small intestines and promote calcium absorption.
Author Response
Thank you very much for your and the reviewers’ thoughtful and constructive comments on our manuscript. We are grateful for the opportunity to clarify and improve our manuscript. Please find our point-by-point responses below:
- Abstract wording clarification: We sincerely appreciate the reviewer’s careful reading. The sentence in the abstract has been revised to eliminate the contradiction.
- Statistical analysis section: Thank you for this helpful suggestion. We have now included a standalone section describing the statistical analyses used in the study.
- Figure 6 improvements: We agree that including statistical significance directly on the bars enhances clarity. Figure 6 has been updated accordingly to show the p-values or significance indicators above the relevant comparisons. We have also now added the statistical significance into the other figures.
- BMMSC uptake suggestion: We appreciate the reviewer’s insight regarding the potential role of vitamin D in mesenchymal stem cells and osteogenesis. While this is an interesting direction, unfortunately due to the lack of current funding and budget cuts, we are unable to pursue additional experiments at this time. However, we have acknowledged this valuable idea as a future direction in the Discussion section.
- Vitamin D receptor and metabolic enzyme expression: The reviewer’s suggestion to explore VDR and CYP24A1 expression in adipocytes was something that we also wanted to do. We were unfortunately unable to carry out additional experiments due to funding cuts that have resulted in the termination of this research project. We have modified our Discussion to include support for the potential involvement of these pathways with relevant literature that justifies the hypothesis of adipocyte involvement in vitamin D metabolism.
We sincerely hope that these revisions and clarifications address the concerns raised and that the current version of the manuscript is suitable for publication. We are thankful for the reviewer’s valuable input.
Reviewer 2 Report
Comments and Suggestions for Authors
The authors have submitted a research article regarding how vitamin D3 and its major metabolites 25-hydroxyvitamin D3 and 1,25-dihydroxyvitamin D3 are taken up by cultured human adipocytes and to what extent they are retained by measuring the levels of the vitamin D3 and the metabolites, illustrating a hypothesis suggesting that the importance of vitamin D3 in human metabolism and pathology can be estimated based on experimental data on the amount of vitamin D3 and its major metabolites retained in adipocytes over time. This issue is of interest, and impact of their results might be strong. My overall concern with the article describing the current available data regarding beneficial availability of the levels of vitamin D3 against various human pathophysiologic status offer something substantial that helps advance our understanding of putative role of vitamin D3 on maintaining a healthy physiological state available in clinic.
To strengthen authors’ perspectives, the authors are strongly recommended to add a “pharmacology” discussion in detail regarding known vitamin D3 effects on proper calcium ion concentrations, for instance. The opposite, toxicological effects of expected outcomes, if known, may influence largely the authors’ perspective. It will be of interest to readers to describe what the results of this study predict about the physiological functions of vitamin D3.
Author Response
We appreciate the reviewer’s thoughtful evaluation and insightful recommendation to strengthen the discussion of our findings by incorporating a broader pharmacological perspective. In response, we have expanded the Discussion section to highlight the physiological relevance of vitamin D₃, particularly its well-established role in calcium homeostasis and its influence on systemic mineral metabolism.
Reviewer 3 Report
Comments and Suggestions for Authors
This is an important study analyzing uptake and intracellular concentrations of vitamin D₃, 25(OH)D₃, and 1,25(OH)₂D₃ over time in human adipocytes in an in vitro setting. I have the following comments:
Please provide the manufacturers and catalog information for the reagents listed in the Materials and Methods section, including the adipogenic cocktail (line 102), lipogenic maintenance medium (line 104), and glycine-based enzyme buffer (line 132). If these reagents were prepared in-house, please provide detailed compositions, including specific doses or concentrations.
Could you please include the following experimental results (either gene or protein expression) in the Results section:
- What type of adipocytes were generated using your differentiation protocol—white or brown?
- Were there any morphological changes in adipocytes after treatment with vitamin D₃ or active vitamin D₃ at a concentration of 10⁻⁶ M?
- Was there any increase in cell death in differentiated human adipose cells following treatment with vitamin D₃ or active vitamin D₃ at 10⁻⁶ M at various time points?
- Were there any changes inCYP24A1 expression before and after treatment with vitamin D₃ or active vitamin D₃ at 10⁻⁶ M across different time points?
- Were there any changes inVDR expression before and after treatment with vitamin D₃ or active vitamin D₃ at 10⁻⁶ M across different time points?
Regarding the Discussion and Conclusion sections, without the inclusion of the additional experimental data listed above and without in vivo context, the conclusions presented in lines 254–256 and 298–299 are not fully convincing.
Author Response
- We appreciate the reviewer’s careful reading. We have now added the manufacturer names and catalog numbers for all commercially obtained reagents, including the adipogenic cocktail (line 102), lipogenic maintenance medium (line 104), and glycine-based enzyme buffer (line 132). For reagents that were prepared in-house, we have included detailed compositions with specific concentrations in the revised Materials and Methods section.
- We appreciate the reviewer’s suggestion.The adipocytes generated using our differentiation protocol are white adipocytes. The information is added into our manuscript.
- We appreciate the reviewer’s suggestion. We have now included a description of adipocyte morphology following treatment with vitamin D₃ and its metabolites.
- We appreciate the reviewer’s insightful comments regarding CYP24A1 and VDR expression changes in response to vitamin D₃ and its active form. While we agree that such data would further strengthen the study, due to current funding constraints and institutional budget cuts, we are unfortunately unable to conduct additional experiments at this time.
Nevertheless, we have revised the Discussion and Conclusion sections (lines 254–256 and 298–299) based on the data we have obtained. We have now included in the Conclusion section wording that qualifies the conclusion and need the further experimentation to confirm the hypothesis.
Round 2
Reviewer 2 Report
Comments and Suggestions for Authors
The authors have done a good job responding to reviewer comments and concerns in their revision. I believe the manuscript is improved as a result. Now I recommend that this revised version of the manuscript can be accepted for publication in Nutrients.
Reviewer 3 Report
Comments and Suggestions for Authors
No further comments.